# Possible Hydrochemical Processes Influencing Dissolved Solids in Surface Water and Groundwater of the Kaidu River Basin, Northwest China

**Dalong Li [1,2], Haiyan Chen [3,*], Shaofeng Jia [1,2] and Aifeng Lv [1,2]**

[1]  Key Laboratory of Water Cycle and Related Land Surface Processes, Institute of Geographic Science and Natural Resources Research, Chinese Academy of Sciences, Beijing 100101, China; lidalong13@mails.ucas.ac.cn (D.L.); jiasf@igsnrr.ac.cn (J.S.); lvaf@163.com (A.L.)

[2]  University of Chinese Academy of Sciences, Beijing 100049, China

[3]  College of Geography and Environmental Science, Hainan Normal University, Haikou 571158, China

*  Correspondence: chenhaiyan@hainnu.edu.cn

**Abstract:** Hydrochemical processes under intense human activities were explored on the basis of the hydrochemical characteristics of 109 surface water samples and 129 groundwater samples collected during August 2015 to September 2016, in the Kaidu River Basin. Results obtained in this study indicated that the water in the basin was neutral to slightly alkaline with low total dissolved solids. Rock weathering and evaporation controlled the natural hydrochemical mechanisms. Mountain groundwater and stream water were dominated by $Ca^{2+}$-$HCO_3^-$ type water, whereas the plains groundwater was dominated by mixed type water. The results of principal component analysis demonstrated that water-rock interaction and human activity explained 71.6% and 12.9% of surface water hydrochemical variations, respectively, and 75.1% and 14.2% of groundwater hydrochemical variations, respectively. Sulfate, chloride, and carbonate weathering were the major water-rock interaction processes. Livestock farming and agricultural activities were the primary human activities influencing the water hydrochemistry. In addition, cation exchange is another important process influencing the hydrochemical characteristics in the study area. This study would be helpful in forecasting of water quality in arid areas.

**Keywords:** stream water; groundwater; Kaidu River Basin; hydrochemical process; water–rock interaction; human activity

---

## 1. Introduction

Arid and semiarid areas have always suffered from water resource shortage. With increasing population and developing economy, overexploitation of water resources and water pollution have been increasing in severity, and therefore water resource shortage has become more serious [1]. Water quality in arid basins is influenced by a various of hydrochemical processes, such as water-rock interaction, land reclamation, wastewater infiltration, sewage exfiltration, and agricultural irrigation [2–4]. Due to the complex influences of multiple natural and anthropogenic processes in arid areas, identifying the major hydrochemical processes is difficult but highly useful [5,6].

Various studies have been conducted to evaluate the effect of natural and anthropogenic processes on hydrochemistry in different environments. Wang et al. [7] studied the hydrochemical characteristics of ground ice in permafrost regions on the Qinghai-Tibet plateau in China; the results indicated that soil moisture, air temperature, and the thickness of the active layer are the major factors controlling variations of soil water chemistry. In the Wiesent River Basin, southern Germany, bedrock geology remains the dominant controlling factor of the major ionic chemistry and agricultural influences

are the strongest near the headwaters [8]. Human activities including groundwater exploitation, agricultural activities, industrial emission and even the construction of Three Gorges Dam were factors that led to the increase in pH, $NO_3$-N, and $Cl^-$ in groundwater from 1992 to 2010, in the Jianghan Plain in China [9]. Shi et al. [3] assessed the effect of seawater intrusion, precipitation infiltration, granite weathering, anthropogenic pollution, and concrete material dissolution to shallow groundwater in a highly urbanized coastal city, Shenzhen, China. The results of principal component analysis (PCA) demonstrated that water-rock interactions and concrete material dissolution are the most important factors affecting shallow groundwater hydrochemistry. Li et al. [10] claimed that halite and carbonate dissolution with strong cation exchange are the major sources of the high salinity in groundwater in the Northwest Namibia. These studies discussed hydrochemical processes under different environments, as well as the effect of these processes on water quality. However, in arid areas, scholars have focused more on water quantity rather than quality [11–14]. In fact, hydrochemical processes, particularly water quality, should receive the same level of attention in arid areas, as they are related to water resource usability.

The Kaidu River is the major recharge source of the Bosten Lake, which is the largest inland freshwater lake in China (with an area of 1210.5 $km^2$ and storage of $73.03 \times 10^8$ $km^3$ when the lake elevation is 1047 m above sea level (a.s.l.)). The river is the major water resource for life and economic development in the Bayingolin Mongol Autonomous Prefecture (Bazhou), as well as for the ecological construction and environmental protection in the Yanqi Basin, the Konqi River Basin, and the lower reaches of the Tarim River Basin. Security of water resources, in the Kaidu River Basin, is of great importance for regional sustainable development. Hydrochemical processes are key factors that influence the security of the water quality. Moreover, in recent years, with the rapid development of irrigation agriculture and animal husbandry, a large amount of residue of pesticide and chemical fertilizer is discharged into soil and water, threatening the security of water resources.

The hydrochemical data of surface water and groundwater, in the Kaidu River Basin, are presented in this study, with the objectives of exploring the predominant mechanisms and hydrochemical processes that control the hydrochemistry of both surface water and groundwater under the influence of intense human activities. This work should be useful for the understanding of water quality changes in arid areas, which is necessary for regional water resource management.

## 2. Study Site

The Kaidu River Basin is located at the southern slope of the Tianshan Mountains and the northern rim of the Tarim River Basin (Figure 1). It covers an area of 44,147 $km^2$, with an altitude ranging from 928 m a.s.l. to 4796 m a.s.l. The Kaidu River originates from glaciers in the Tianshan Mountains, runs through the small Yourdusi Basin, the large Yourdusi Basin and, then, along the mountain valley to the Dashankou hydrometric station, through the Yanqi Basin, and finally flows into the Bosten Lake. It is one of the largest rivers on the southern slope of the Tianshans (with a mean annual discharge of 108.42 $m^3$/s) [15].

The study area is situated in central Eurasia, controlled by a typical temperate continental climate. Climate characteristics differ significantly between the mountains and piedmont plains. The average temperatures are −4.2 °C and 8.9 °C with average precipitation of 280.5 mm and 84.3 mm, in the mountains and the plains, respectively. Approximately 86% and 80% of annual precipitation in the mountains and the plains, respectively, is concentrated in the May to September period.

The Kaidu River Basin is located at the southern Tianshan geosynclinal folded zone; folds and faults have developed under this influence. The North-West-West (NWW) and Nort-East-East (NEE) trending faults are the major two groups of faults; the NWW trending fault is in accordance with the main tectonic line. The exposure strata are mainly Cenozoic-Quaternary sediments in the small Yourdusi Basin, large Yourdusi Basin, and Yanqi Basin; Paleozoic sediments in the alpine valley region; and Proterozoic era sediments in the valley area at the east of the small Yourdusi Basin. The outcropped lithologies in the Kaidu River Basin are mainly glutenite, argillaceous sandstone, mudstone, clay, and

silt, which are rich in carbonate, sulfate, and halite [16,17]. The groundwater resource is richer in the mountainous areas than in the plains.

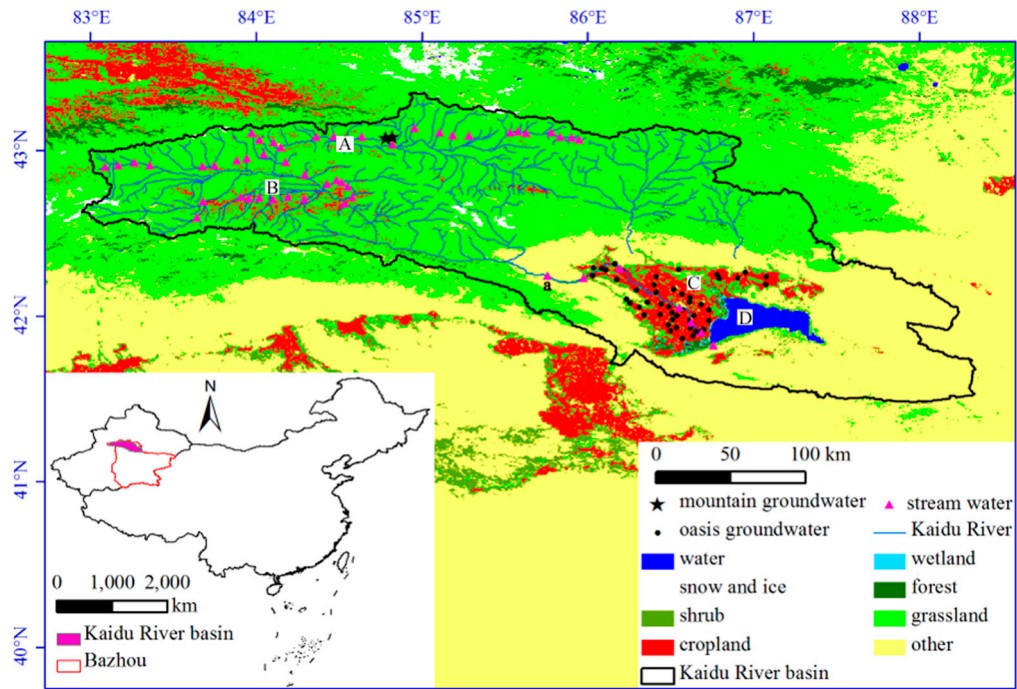

**Figure 1.** Geographic location and land cover of study area and sampling stations. (**A**) small Yourdusi Basin; (**B**) large Yourdusi Basin; (**C**) Yanqi Basin; (**D**) Bosten Lake; (**a**) Dashankou station.

The annual river runoff at the mountain outlet station ranges from $24.61 \times 10^8$ m$^3$ to $57.26 \times 10^8$ m$^3$, with a mean value of $35.32 \times 10^8$ m$^3$. The maximum runoff occurred in July and August. The spatial and temporal variations of groundwater depth in the downstream of the Kaidu River Basin are very significant. The groundwater depth increases with the distance from the Kaidu River. The annual mean groundwater level ranges from 3 m to more than 28 m below the ground. In recent years, the groundwater level has decreased significantly, averaging a 5 m to 14 m decrease. Because of over pumping for irrigation, the groundwater level is the shallowest in winter and deepest during the growing season (June to August) and during the autumn irrigation season (October). In regions with large farmland areas, the groundwater level is deeper than in other areas.

The major land uses in the basins are grasslands (accounting for 61.60% of the total study area), deserts (27.72%), and farmland (6.07%) (Figure 1). In the mountains, the major soil types are tundra soil, meadow soil, steppe soil, chestnut soil, and brown desert soil. In the plains, the major soil types are moisture soil, brown desert soil, saline soil, and irrigation desert soil [16,17].

## 3. Data and Methods

### 3.1. Sampling

In order to avoid the effects of weather, clear weather (zero precipitation) for at least three consecutive days was required before sampling. From August 2015 to September 2016, five systematic sampling campaigns in four seasons (August 2015, January 2016, April 2016, August 2016, and September 2016) were conducted (Figure 1). The meteorological seasons are referred to as spring (March, April, and May), summer (June, July, and August), autumn (September, October, and November), and winter (December, January, and February). A total of 106 stream water samples, 121 groundwater samples, and 8 mountain groundwater samples were collected.

The stream water samples were collected from the middle of the river's cross-section. Groundwater samples were collected from farm wells or domestic water wells along the river. Human activities have the greatest influence on shallow groundwater. Therefore, we only collected shallow groundwater from well depths less than 50 m. The well depth was recorded; however, if the wells were closed, we did not measure the groundwater table. The wells were purged at least half an hour before sampling. All water samples were collected in three 500 mL clean and dry high-density polyethylene plastic bottles. Prior to sampling, the bottles were soaked for 24 h in deionized water in the laboratory. After soaking, the bottles were rinsed three times with deionized water and, then, dried at room temperature. All bottles were washed three times again using the sample water in situ before sampling and, then, filled with sample water and sealed immediately using parafilm to prevent contamination. After collection, samples were transported immediately via a cooling box.

Analyses were conducted in the Central Laboratory of Xinjiang Institute of Ecology and Geography, Chinese Academy of Sciences immediately after returning from the field. The pH was measured using a pH meter (PHJS-4A, Inesa Inc., Shanghai, China). Major cationic and anionic analyses were conducted by ion chromatography systems (Dionex ICS-5000, Sunnyvale, CA, USA). The total dissolved solids (TDS) was measured using a weighing method, with a precision within 1 μg/L.

### 3.2. Saturation Indices (SIs)

As an important parameter in geochemistry and hydrogeology, the saturation index is used to identify whether a water sample tends to dissolve or precipitate a certain kind of mineral [18]. In this study, we calculated the saturation indices (SIs) of anhydrite, aragonite, calcite, dolomite, gypsum, and halite using the following equation [19]:

$$SI = \log\left(\frac{IAP}{k_s(T)}\right) \tag{1}$$

where IAP is the relevant activity product of the solution, $k_s(T)$ is the equilibrium constant of the reaction considered at the sample temperature (°C). The SIs were calculated using the geochemical computer model PHREEQC [20]. When the water is saturated with the dissolved mineral, the SI equals zero; positive values indicate oversaturation, while negative values indicate undersaturation [21,22].

### 3.3. Cation Exchange

The chloro-alkaline indices (CAI-I and CAI-II), proposed by Schoeller [23], are frequently used to explore the cation exchange between water and its host environment during water migration. The CAI-I and CAI-II are calculated by the following equations:

$$CAI - I = \frac{Cl^- - \left(Na^+ + K^+\right)}{Cl^-} \tag{2}$$

$$CAI - II = \frac{Cl^- - \left(Na^+ + K^+\right)}{HCO_3^- + SO_4^{2-} + CO_3^{2-}} \tag{3}$$

Negative Schoeller indices indicate $Ca^{2+}$ or $Mg^{2+}$ exchange in solution with $Na^+$ or $K^+$, i.e., $Ca^{2+}$ or $Mg^{2+}$ are removed from solution and $Na^+$ or $K^+$ are released into it. The unit of the ion content is denoted in meq/L. Negative values indicate chloro-alkaline disequilibrium. This reaction is known as the cation exchange reaction; the host rocks are the primary sources of dissolved solids in the water during this process, while the positive Scholler indices indicate the reverse ion exchange.

## 4. Results

### 4.1. Water Hydrochemistry

Water in the basin was neutral to slightly alkaline with the pH varying from 6.82 to 8.30 (mean 7.95). The TDS values (Figure 2a) increased along the water flow direction, indicating more soluble matter downstream of the basin [24,25]. The mean TDS ranged from 130.92 mg/L (mountain groundwater) to 422.58 mg/L (plain groundwater) (Table 1). It is lower than the drinking water standard of WHO [26,27].

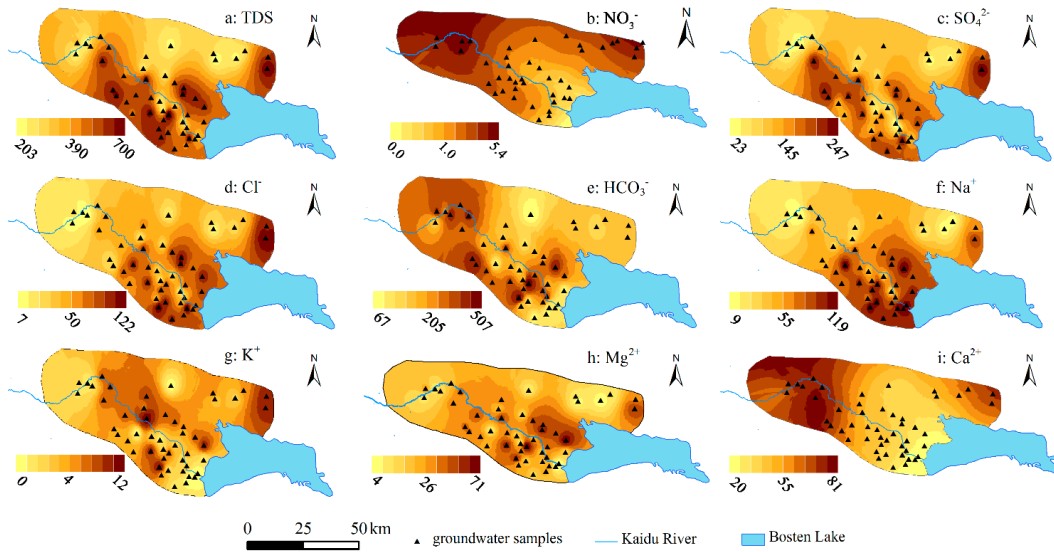

**Figure 2.** Spacial distributions of TDS and the major ions in plain groundwater. The unit is mg/L.

**Table 1.** Statistics of the stream water and groundwater hydrochemistry and Schoeller indices.

| Location | Water Type | | $HCO_3^-$ (mg/L) | $Cl^-$ (mg/L) | $SO_4^{2-}$ (mg/L) | $NO_3^-$ (mg/L) | $Ca^{2+}$ (mg/L) | $Mg^{2+}$ (mg/L) | $K^+$ (mg/L) | $Na^+$ (mg/L) | TDS (mg/L) | pH | CAI-I | CAI-II |
|---|---|---|---|---|---|---|---|---|---|---|---|---|---|---|
| mountain | stream water | mean | 136.29 | 7.33 | 28.68 | 0.70 | 44.72 | 8.44 | 1.37 | 8.01 | 166.69 | 8.04 | −1.22 | −0.06 |
| | | min | 42.29 | 0.73 | 0.00 | 0.03 | 13.79 | 0.94 | 0.21 | 0.74 | 44.99 | 7.56 | −4.96 | −0.39 |
| | | max | 315.78 | 36.27 | 174.69 | 1.60 | 92.17 | 41.67 | 4.59 | 36.10 | 431.53 | 8.29 | 0.53 | 0.11 |
| | groundwater | mean | 116.33 | 7.96 | 11.92 | 0.43 | 36.09 | 7.14 | 1.37 | 8.27 | 130.91 | 8.05 | −1.22 | −0.10 |
| | | min | 65.10 | 2.56 | 0.00 | 0.17 | 24.06 | 4.98 | 0.10 | 4.12 | 84.09 | 7.93 | −2.29 | −0.37 |
| | | max | 160.17 | 16.63 | 23.09 | 0.64 | 48.11 | 8.80 | 4.07 | 21.29 | 158.39 | 8.25 | 0.40 | 0.08 |
| plain | stream water | mean | 176.87 | 14.43 | 47.26 | 0.80 | 51.18 | 15.70 | 2.33 | 19.36 | 238.69 | 8.07 | −1.07 | −0.13 |
| | | min | 117.19 | 6.26 | 31.44 | 0.16 | 43.30 | 10.79 | 0.15 | 6.90 | 193.37 | 7.77 | −2.29 | −0.37 |
| | | max | 249.34 | 35.34 | 92.38 | 1.80 | 65.44 | 23.48 | 4.31 | 53.06 | 335.35 | 8.30 | −0.59 | −0.04 |
| | groundwater | mean | 203.85 | 57.52 | 118.93 | 2.25 | 51.56 | 27.46 | 3.81 | 61.39 | 422.58 | 7.91 | −1.15 | −0.26 |
| | | min | 67.71 | 7.65 | 19.74 | 0.00 | 12.52 | 4.56 | 0.59 | 7.27 | 194.89 | 7.27 | −3.04 | -0.75 |
| | | max | 656.37 | 185.97 | 346.41 | 10.13 | 153.01 | 117.39 | 25.03 | 201.24 | 948.19 | 8.28 | 0.24 | 0.04 |

The mean ion concentrations were the lowest in mountain groundwater and the highest in plain groundwater. $HCO_3^-$ and $Ca^{2+}$ were the dominant ions; the mean concentration of $HCO_3^-$ ranges from 116.33 to 203.85 mg/L. The mean concentration of $Ca^{2+}$ ranges from 36.09 to 51.56 mg/L, while the mean concentration of $SO_4^{2-}$ ranges from 11.92 to 118.93 mg/L. The mean concentration of $Mg^{2+}$ ranges from 7.14 to 27.46 mg/L, whereas those of $NO_3^-$ and $K^+$ were the lowest. The mean concentration of $NO_3^-$ ranges from 0.43 to 2.25 mg/L, while the mean concentration of $K^+$ ranges from 1.37 to 3.81 mg/L.

The contributions of different ions to total ions varied with water types. In stream water, $Ca^{2+}$ accounted for 65% of total cations (based on meq $L^{-1}$ values), followed by $Mg^{2+}$ (22%), $Na^+$ (12%), and $K^+$ (1%), and anions by $HCO_3^-$ (71%) > $SO_4^{2-}$ (21%) > $Cl^-$ (8%). They were very similar to the major ions of the mountain groundwater, with cations as $Ca^{2+}$ (63%) > $Mg^{2+}$ (19%) > $Na^+$ (17%) > $K^+$ (1%) and anions as $HCO_3^-$ (76%) > $SO_4^{2-}$ (13%) > $Cl^-$ (11%). The order of cation concentration is in accordance with that of standard crustal materials ($Ca^{2+}$ > $Mg^{2+}$ > $Na^+$ > $K^+$) [28]. Cations followed

the order of Ca$^{2+}$ (35%) = Na$^+$ (35%) > Mg$^{2+}$ (29%) > K$^+$ (1%) and anions of HCO$_3^-$ (46%) > SO$_4^{2-}$ (33%) > Cl$^-$ (21%) for the plain groundwater (Table 1). The contribution of NO$_3^-$ to total anions was less than 1% in the Kaidu River Basin (Table 1).

The predominant hydrochemical types of stream water are Ca$^{2+}$-HCO$_3^-$ type water followed by mixed type water. The hydrochemical types of mountain groundwater are very similar to stream water. The predominant hydrochemical types of the plain groundwater are also Ca$^{2+}$-HCO$_3^-$ and mixed type water. The Na$^+$, Cl$^-$, and SO$_4^{2-}$ counted as a much larger proportion for groundwater hydrochemistry than for surface water. The water types of both stream water and groundwater in the plain did not change significantly with seasons (Figure 3).

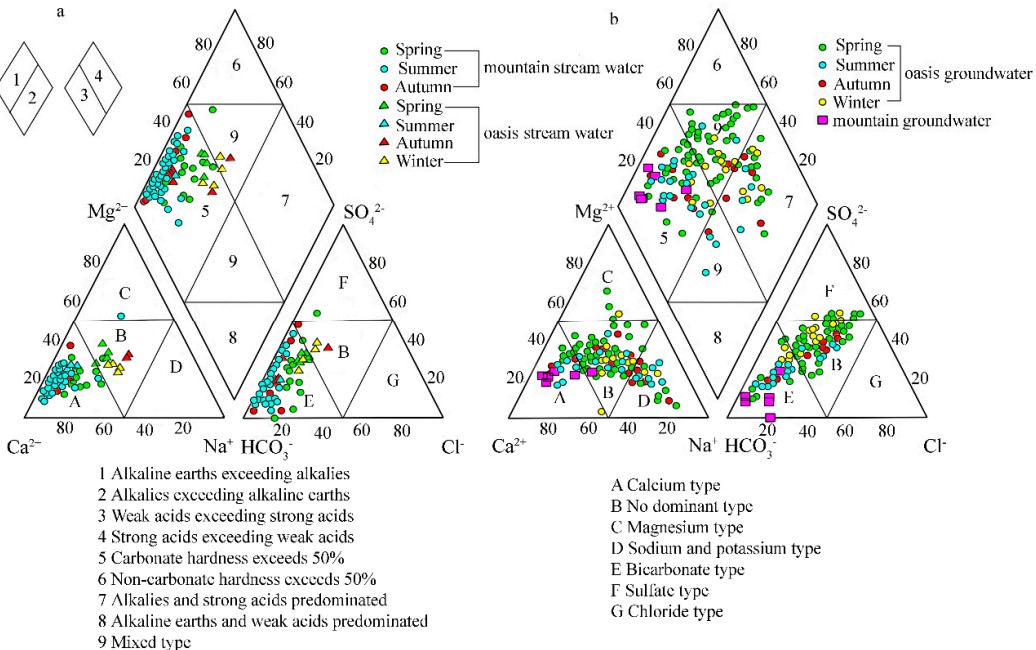

**Figure 3.** Piper diagrams of (**a**) stream water and (**b**) groundwater in different seasons.

Though there are some differences, the seasonal variations of the major chemical ions in the stream water and the plain groundwater are very similar, except for Na$^+$ and Cl$^-$ (Figure 4). Owing to the wide distribution of halite, the concentration of Na$^+$ and Cl$^-$ in groundwater is high all year round [29].

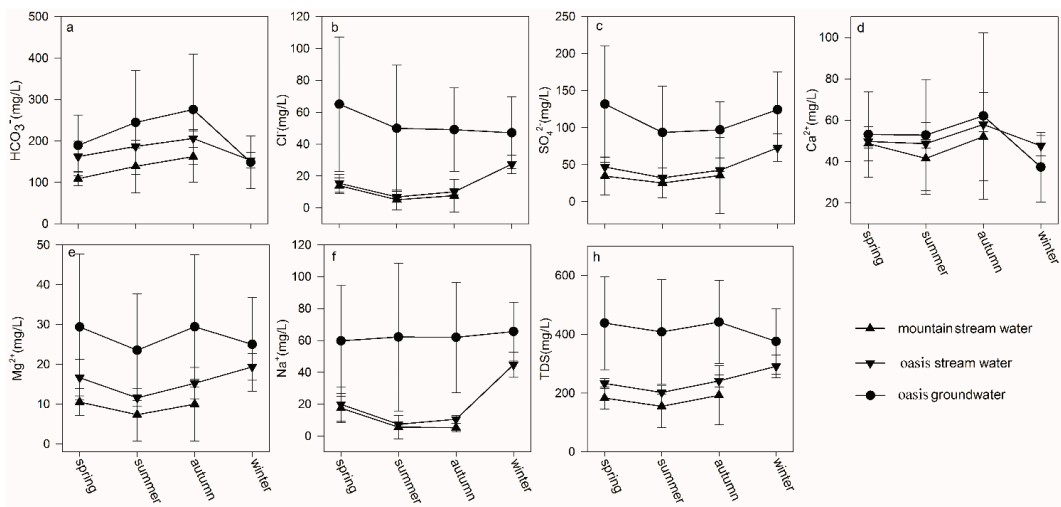

**Figure 4.** Seasonal variation of (**a**–**e**) major chemistry ions and (**h**) TDS of stream water and groundwater.

Figure 2 displayed the spatial distributions of the major ions of groundwater in the plains. The $SO_4^{2-}$, $Na^+$, and $Cl^-$ concentrations were much higher in the southern and the eastern part of the plain. The concentrations of $HCO_3^-$ and $Ca^{2+}$ decreased along the water flow path. This indicates that the water in the mountain has a stronger dissolving ability for carbonate as compared with water in the plain in the downstream of the Kaidu River. The influence of evaporation is stronger on water in plains as compared with water in the mountain areas.

### 4.2. Water-Rock Interaction

The dominant hydrochemical formation mechanism (including precipitation dominance, rock weathering dominance, or evaporation dominance) is determined according to the ratios of $Na^+/(Na^+ + Ca^{2+})$ against TDS [30]. In stream water and mountain groundwater, the average values of $Na^+/(Na^++Ca^{2+})$ is lower than 0.5, and the TDS values are 188.85 and 130.91 mg/L, respectively (Figure 5), showing that rock weathering is the dominant hydrochemical mechanism. In plain groundwater, the average value of $Na^+/(Na^+ + Ca^{2+})$ is 0.64, which is much higher than 0.5, and the TDS is 461.30 mg/L, both values are much higher than that of stream water and mountain groundwater (Figure 5). This indicates that the hydrochemistry of groundwater is not only controlled by rock weathering but also by evaporation processes.

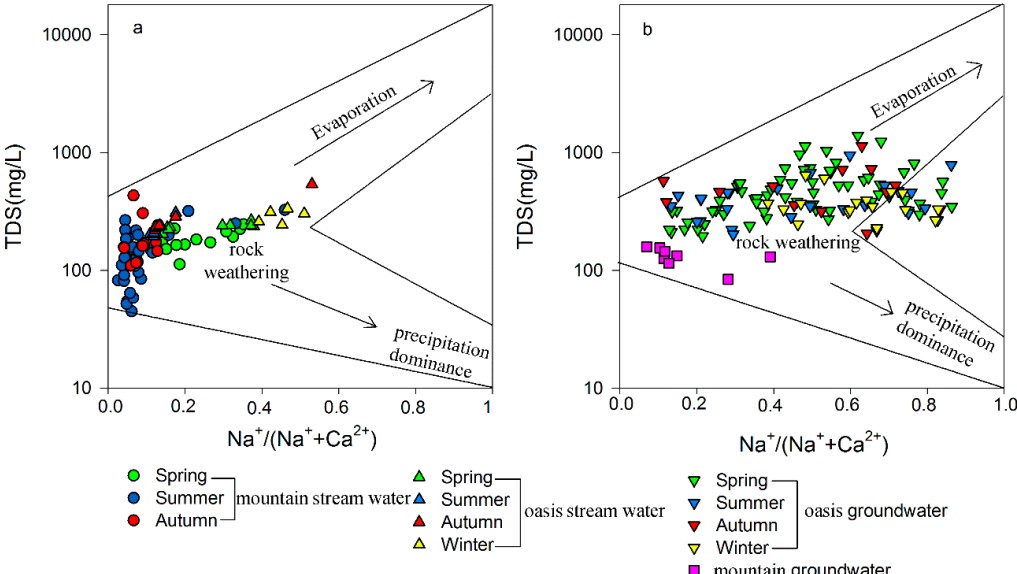

**Figure 5.** Gibbs diagram of (**a**) stream water and (**b**) groundwater.

Bivariate plots are effective tools to reflect the major water–rock interaction processes [3,25]. In this study, all water samples were plotted around the 1:1 line in the $(Ca^{2+} + Mg^{2+})$ vs. $(HCO_3^- + SO_4^{2-})$ plot (Figure 6a). Moreover, the carbonate minerals and sulfate minerals are widely distributed in the area. This indicates that the carbonate and sulfate weathering, similar to dolomite and gypsum weathering, are important geochemical processes [31].

Correlations among the major ions in stream water and groundwater are shown in Table 2. Most of the major cations and anions in stream water and plain groundwater were significantly correlated with TDS, indicating that these ions are dissolved continuously, resulting in the rise of TDS [24]. However, for mountain groundwater, only $Ca^{2+}$ and $HCO_3^-$ were significantly correlated with TDS. This is due to the high proportion of $Ca^{2+} + HCO_3^-$ (71%) to the total ions, indicating that carbonate weathering is the major hydrochemical process in mountain groundwater.

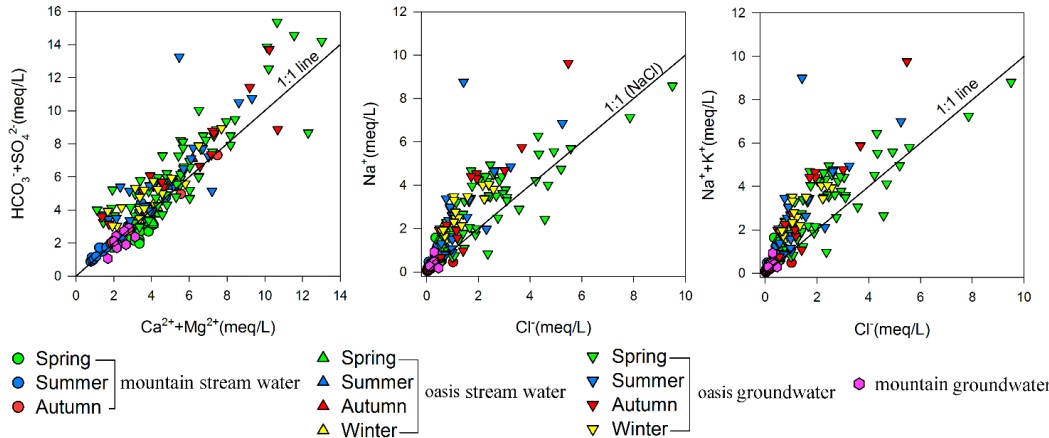

**Figure 6.** Scatter diagrams showing relationship between (**a**) $Ca^{2+}$ + $Mg^{2+}$ vs. $HCO_3^-$+$SO_4^{2-}$; (**b**) $HCO_3^-$ vs. $Ca^{2+}$; (**c**) $HCO_3^-$ vs. $Mg^{2+}$; (**d**) $SO_4^{2-}$ vs. $Ca^{2+}$; (**e**) $SO_4^{2-}$ vs. $Mg^{2+}$; (**f**) $SO_4^{2-}$ vs. $Na^+$; (**g**) $Cl^-$ vs. $Ca^{2+}$; (**h**) $Cl^-$ vs. $Mg^{2+}$; and (**i**) $Cl^-$ vs. ($Na^+$ + $K^+$) of stream water and groundwater in different seasons.

**Table 2.** Correlation matrix for the major parameters in stream water and groundwater.

| | $HCO_3^-$ | $Cl^-$ | $SO_4^{2-}$ | $Ca^{2+}$ | $Mg^{2+}$ | $K^+$ | $Na^+$ | TDS |
|---|---|---|---|---|---|---|---|---|
| **Stream water** | | | | | | | | |
| $HCO_3^-$ | 1 | 0.403 ** | 0.217 * | 0.735 ** | 0.578 ** | 0.168 | 0.261 * | 0.720 ** |
| $Cl^-$ | | 1 | 0.523 ** | 0.425 ** | 0.699 ** | 0.717 ** | 0.871 ** | 0.775 ** |
| $SO_4^{2-}$ | | | 1 | 0.469 ** | 0.731 ** | 0.410 ** | 0.581 ** | 0.775 ** |
| $Ca^{2+}$ | | | | 1 | 0.500 ** | 0.359 ** | 0.275 ** | 0.770 ** |
| $Mg^{2+}$ | | | | | 1 | 0.582 ** | 0.716 ** | 0.883 ** |
| $K^+$ | | | | | | 1 | 0.662 ** | 0.558 ** |
| $Na^+$ | | | | | | | 1 | 0.725 ** |
| TDS | | | | | | | | 1 |
| **Plain groundwater** | | | | | | | | |
| $HCO_3^-$ | 1 | −0.051 | −0.012 | 0.470 ** | 0.447 ** | 0.348 ** | 0.012 | 0.307 ** |
| $Cl^-$ | | 1 | 0.913 ** | 0.310 ** | 0.695 ** | 0.327 ** | 0.773 ** | 0.903 ** |
| $SO_4^{2-}$ | | | 1 | 0.399 ** | 0.681 ** | 0.373 ** | 0.724 ** | 0.931 ** |
| $Ca^{2+}$ | | | | 1 | 0.382 ** | 0.311 ** | −0.027 | 0.516 ** |
| $Mg^{2+}$ | | | | | 1 | 0.521 ** | 0.459 ** | 0.811 ** |
| $K^+$ | | | | | | 1 | 0.184 * | 0.461 ** |
| $Na^+$ | | | | | | | 1 | 0.754 ** |
| TDS | | | | | | | | 1 |
| **Mountain groundwater** | | | | | | | | |
| $HCO_3^-$ | 1 | −0.442 | 0.143 | 0.780 * | −0.029 | 0.197 | −0.701 | 0.754 * |
| $Cl^-$ | | 1 | −0.064 | 0.194 | 0.712 * | 0.690 | 0.239 | 0.097 |
| $SO_4^{2-}$ | | | 1 | 0.192 | 0.447 | −0.084 | 0.456 | 0.544 |
| $Ca^{2+}$ | | | | 1 | 0.477 | 0.713 * | −0.592 | 0.912 ** |
| $Mg^{2+}$ | | | | | 1 | 0.539 | 0.341 | 0.589 |
| $K^+$ | | | | | | 1 | −0.451 | 0.493 |
| $Na^+$ | | | | | | | 1 | −0.291 |
| TDS | | | | | | | | 1 |

* Indicates that the correlation is significant at 0.05 level; ** indicates that the correlation is significant at 0.01 level.

The high positive correlation between $Ca^{2+}$ and $HCO_3^-$ in stream water and mountain groundwater indicates calcite weathering, as expressed in Reaction (1) [24]. The high correlations between $Mg^{2+}$ and $HCO_3^-$, $Ca^{2+}$ and $HCO_3^-$, and $Ca^{2+}$ and $Mg^{2+}$ in stream water indicate the weathering of dolomite, as expressed in Reaction (2) [32]. The high correlations between $Mg^{2+}$ and $SO_4^{2-}$, and $Na^+$ and $SO_4^{2-}$ in stream water and mountain groundwater indicate the dissolution of sulfate [33]. The high correlations

between $Na^+$ and $Cl^-$ in stream water and plain groundwater, $K^+$ and $Cl^-$, and $K^+$ and $Na^+$ in stream water indicate the dissolution of evaporites [34]. The high correlation between $Mg^{2+}$ and $Cl^-$ in stream water and plain groundwater indicates dissolution of magnesium chloride. The wide distribution of saline soil in the plains offers the source of magnesium chloride [17]. The high correlation between $Mg^{2+}$ and $Na^+$ in stream water indicates the weathering of certain kinds of minerals that contain these two ions, such as anorthose, as shown in Reaction (3) [35]. The high correlation between $K^+$ and $Mg^{2+}$ in stream water indicates biotite weathering, as expressed in Reaction (4) [31].

$$CaCO_3 + CO_2 + H_2O \Leftrightarrow Ca^{2+} + 2HCO_3^- \quad \text{Reaction 1}$$

$$CaMg(CO_3)_2 + 2CO_2 + 2H_2O \Leftrightarrow Ca^{2+} + Mg^{2+} + 4HCO_3^-$$

Dolomite Reaction 2

$$Na_6Ca_4Al_{14}Si_{26}O_{80} + 31H_2O + 14H_2CO_3 \rightarrow 7AlSi_2O_5(OH)_4 + 6Na^+ + 4Ca^{2+} + 14HCO_3^- + 12H_4SiO_4$$

Anorthose kaolinite orthosilicic acid Reaction 3

$$14KAlMg_3Si_3O_{10}(OH)_2 + 98H_2CO_3 + 7H_2O \rightarrow 7Al_2Si_2O_5(OH)_4 + 42Mg^{2+} + 14K^+ + 98HCO_3^- + 28H_4SiO_4$$

Biotite kaolinite orthosilicic acid Reaction 4

The calculated SI values of aragonite, calcite, and dolomite ranged from −0.97 to 0.72, −0.81 to 0.87, and −2.53 to 1.64, with an average of −0.04, 0.12, and −0.18, respectively (Figure 7), indicating that the aragonite, calcite, and dolomite were saturated. They would precipitate under conditions of carbon dioxide content decrease or temperature decrease. However, the SI values of anhydrite, gypsum, and halite ranged from −3.80 to −1.20, −3.55 to −0.95, and −5.94 to −10.67, with an average of −2.23, −1.98, and −7.91 (Figure 7), respectively, indicating the undersaturation of these minerals. They dissolve continually under suitable conditions. In basins in the watershed, including the small Yourdusi Basin, the large Yourdusi Basin, and the Yanqi Basin, lithology is mainly controlled by characteristics of the quaternary sediment. There is a wide distribution of carbonate minerals and sulphate minerals, such as aragonite, calcite, dolomite, anhydrite, and gypsum [36]. Moreover, in the Yanqi Basin, controlled by dry climate and extensive evaporation, there is widespread halite in the topsoil [34].

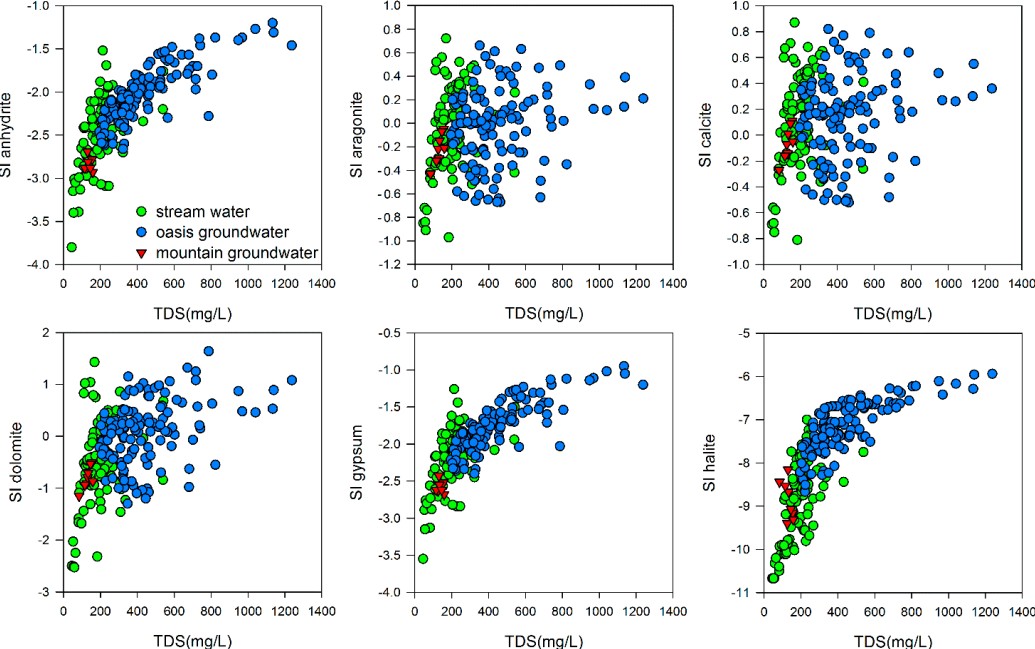

**Figure 7.** Saturation indices (SI) of some minerals versus TDS (mg/L) in stream water and groundwater.

According to the $Na^+$ vs. $Cl^-$ plot and $(Na^+ + K^+)$ vs. $Cl^-$ plot, though stream water and mountain groundwater samples were plotted very close to the 1:1 line, most samples were scattered above the 1:1 line (Figure 6), indicating other $Na^+$ sources, except for halite, dissolution [34]. The cation exchange between $Ca^{2+}$ or $Mg^{2+}$ and $Na^+$ or $K^+$ helps explain the relatively high $Na^+$ concentration in water [33]. In the Kaidu River Basin, the average values of Scholler indices of stream water and groundwater were negative (Table 1), indicating that ion exchange is an important water–rock interaction process [24]. In this process, $Ca^{2+}$ or $Mg^{2+}$ was removed from solution and $Na^+$ was released into it (Reaction (5)).

$$Ca^{2+}/Mg^{2+} + Na - X \rightarrow Na^+ - Ca/Mg + X \quad \text{Reaction 5}$$

### 4.3. Effect of Human Activity

$NO_3^-$ concentration is an indicator of intensity of human activity [37,38]. The $NO_3^-$ concentration in plain groundwater ranged from 0 to 5.4 mg/L and decreased from west to east, in the spring of 2016 (Figure 2b). The $NO_3^-$ concentration was lower than 2 mg/L in surface water, while it was higher in pastoral areas with altitudes ranging from 1400 to 3100 m a.s.l. as compared with the high mountains where the altitude is higher than 3100 m or the arable land where the altitude is lower than 1400 m (Figure 8). This indicates that $NO_3^-$ is mainly obtained from livestock farming in the mountainous area and irrigation in the plains. A large amount of nitric oxides discharge into the environment due to animal excrement and fertilization. The mountain area is the major water gathering area, as well as the major pastoral region in the basin. The plain in the downstream of the river is dominated by irrigation agriculture, which is heavily dependent on drip irrigation. The soil water content is very low in the study area, therefore agricultural waste is not be immediately discharged into groundwater or river. Thus, livestock farming has a larger impact on $NO_3^-$ concentration than irrigation agriculture.

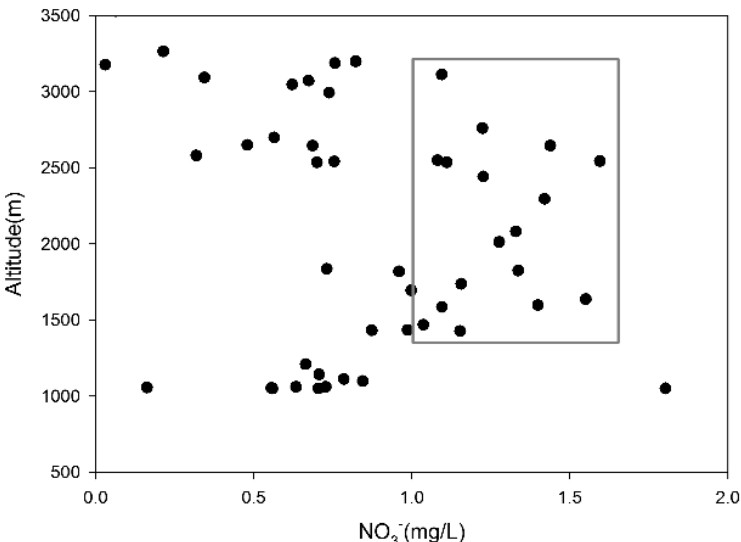

**Figure 8.** Altitude versus $NO_3^-$ concentrations of the surface water.

The population of Yanqi oasis had increased before 2010 and then decreased. The population was 432,369, in 2005, increased to 475,178, in 2010, then decreased to 443,998, in 2015. The farmland area, orchard area, number of livestock, and the use of pesticides and chemical fertilizer have increased significantly. The sown area increased from 93,520 ha, in 2005, to 121,246.7 ha, in 2015. The use of chemical fertilizer increased from 24,072 ton, in 2005, to 52,051 ton, in 2015. The pesticide use increased from 39,088 kg, in 2005, to 405,604 kg, in 2015. The population of livestock increased from 325,640 sheep units, in 2005, to 404,843,0 sheep units, in 2015 (Table 3). Pesticides and fertilizer residuals in soil did not discharge into groundwater or rivers in arid areas without extreme events such as heavy rainstorms;

however, they were a potential menace for regional water resource security [4,39]. In addition, animal husbandry which was mainly distributed in water source areas directly contaminated the water.

**Table 3.** Population and husbandry development during the recent 10 years, in the Yanqi Oasis.

| Year | Population | Sown Area (ha) | Irrigated Area (ha) | Orchard (ha) | Net Content(t) | | | | Pesticide(kg) | Livestock (Sheep Unit) |
|---|---|---|---|---|---|---|---|---|---|---|
| | | | | | Nitrogen | P Fertilizer | Potassic Fertilizer | Compound Fertilizer | | |
| 2005 | 432,369 | 93,520 | 69,260 | 11,420 | 15,507 | 5929 | 450 | 2186 | 39,088 | 3,256,460 |
| 2010 | 475,178 | 116,213.3 | 92,600 | 18,161.2 | 20,772 | 12,556 | 2680 | 5194 | 293,729 | 2,282,150 |
| 2015 | 443,998 | 121,246.7 | 104,760 | 23,364 | 26,109 | 15,860 | 3998 | 6084 | 405,604 | 4,048,430 |

The data were obtained from the Xinjiang statistical yearbooks of 2006, 2011, and 2015.

### 4.4. Estimation of the Contribution Sources

PCA was conducted to evaluate the contributions of the above discussed factors to water hydrochemistry, in the Kaidu River Basin. Table 4 shows the PCA results of stream water and groundwater.

**Table 4.** Results of principal component analysis for major ions in stream water and groundwater.

| | Factor 1 | | Factor 2 | | Factor 3 | |
|---|---|---|---|---|---|---|
| | SW | GW | SW | GW | SW | GW |
| $Ca^{2+}$ | 0.29 | 0.32 | 0.87 | 0.79 | −0.06 | 0.08 |
| $Mg^{2+}$ | 0.82 | 0.85 | 0.45 | 0.46 | 0.02 | 0.01 |
| $Na^+$ | 0.9 | 0.84 | 0.25 | −0.3 | 0.17 | 0.05 |
| $K^+$ | 0.8 | 0.27 | 0.19 | 0.3 | 0.05 | 0.81 |
| $HCO_3^-$ | 0.23 | −0.03 | 0.93 | 0.95 | 0.02 | 0.22 |
| $SO_4^{2-}$ | 0.8 | 0.96 | 0.12 | −0.03 | −0.08 | 0.12 |
| $Cl^-$ | 0.91 | 0.99 | 0.26 | −0.03 | 0.11 | 0.01 |
| $NO_3^-$ | 0.08 | −0.09 | −0.03 | 0.03 | 0.99 | 0.93 |
| % Variance | 46.66 | 55.29 | 24.98 | 19.85 | 12.87 | 14.16 |
| % Cumulative | 46.66 | 55.29 | 71.64 | 75.14 | 84.51 | 89.3 |
| Source | Sulfate and chloride | | Carbonate | | Human activity | |

SW, stream water and mountain groundwater; GW, plain groundwater.

For surface water, the first three major factors accounted for 84.5% of the total hydrochemical variance. The first factor explained 46.7% of the total variance with high loadings of $SO_4^{2-}$, $Cl^-$, $Na^+$, $K^+$, and $Mg^{2+}$, attributed to sulfate and chloride weathering, such as anhydrite, gypsum, and halite. The second factor accounted for 25.0% of the total variance with high loadings of $HCO_3^-$, $Ca^{2+}$, and $Mg^{2+}$, owing to carbonate weathering, such as aragonite, calcite, and dolomite. The third factor accounted for 12.9% of the total variance with high loadings of $NO_3^-$. It represented an anthropogenic pollution source such as the widely distributed livestock farming in the mountain.

For the plain groundwater, the first three major factors accounted for 89.3% of the total variance. The first factor accounted for 55.3% of the total variance with high loadings of $SO_4^{2-}$, $Cl^-$, $Na^+$, and $Mg^{2+}$, indicating the weathering of sulfate and chloride, such as anhydrite, gypsum and halite. The second factor explained 19.9% of the total variance with high loadings of $HCO_3^-$, $Ca^{2+}$, and $Mg^{2+}$, indicating carbonate weathering, such as aragonite, calcite, and dolomite. The third factor accounted for 14.2% of the total variance with high loadings of $NO_3^-$ and $K^+$, indicating anthropogenic pollution sources including agriculture activities and livestock farming in the plains.

## 5. Discussion

Hydrochemical processes have a direct effect on regional water quality. Factors that influence hydrochemical processes include both natural and human factors. Natural factors include bed rock properties, geomorphological features, and climatic features. Studies at the Hongfeng Lake in

Guizhou province, China found that non-point source pollution caused by heavy rainfall results in increasing the total nitrogen content of the lake [40]. Studies in mountain catchments in northern Taiwan found that the hydrochemical processes during typhoon period are significantly different from other times, the high predictability of ion input and output budgets using stream discharge during the non-typhoon period largely disappeared during the typhoon periods [41]. Chiogna et al. [42] found that hydrochemical processes are significantly affected by drought. For groundwater in the Northwest Namibia, Li et al. [10] found that the TDS increase in groundwater is primarily due to mineral dissolution, rather than evaporation.

The Kaidu River Basin, which is located in an arid area, is characterized by poor precipitation, frequent extreme climatic events, intense evaporation, and a wide distribution of different types of dissoluble minerals. The hydrochemical processes in the basin are affected by bedrock interaction, evaporation, ion exchange, and extreme climatic events. For example, the TDS in surface water and shallow groundwater increased significantly after heavy rainfall.

Human factors that influence hydrochemical processes are mainly related to the types and intensities of human activities. Approximately 27% of the TDS in the Taizi River (Northeast China) resulted from anthropogenic inputs [43]. Base flow solute concentrations (particularly sulfate, chloride, bicarbonate alkalinity, and sodium) in the Chattahoochee River Basin (Georgia, USA) increased with the degree of urbanization [44]. Overuse of chemical fertilizers, industrial sewage discharging, and the vulnerability of groundwater systems to contamination resulted in the $NO_3^-$ contamination of stream water and groundwater in the Hutuo River Basin, northern China [45]. With widely distributed irrigation agriculture, on the one hand, a large amount of pesticide and chemical fertilizer discharges into soil; on the other hand, irrigation, particularly drip irrigation, results in salts concentration in the topsoil [39,46]. Livestock farming, concentrated farming, or nomadism discharge large amounts of nitrogen, phosphorus, potassium, bacteria, methane, ammonia, and other materials into environment [47,48].

Nomadic herding is highly developed in the mountainous areas of the study site, whereas irrigation agriculture is widely distributed downstream of the catchment. In recent years, the population of livestock, irrigation area, and amount of pesticide and fertilizer used have increased significantly (Table 3). A significant number of chemical elements such as nitrogen, phosphorous, and potassium are discharged into soil and water. These change the hydrochemical processes, threaten regional water source quality, and enhance the spatial and temporal variance of hydrochemical characters. For example, the TDS of surface water and shallow groundwater increased significantly after autumn irrigation.

## 6. Conclusions

The major hydrochemical processes, in the Kaidu River Basin, were assessed based on hydrochemical data of stream water and groundwater samples. The predominant hydrochemical water type in stream water and mountain groundwater was $Ca^{2+}$-$HCO_3^-$, while mixed type water predominated the basin groundwater. Groundwater showed the highest TDS values. Owing to the frequent exchange between surface water and groundwater, there was no significant difference between the solute concentrations in stream water and mountain groundwater. Rock weathering and evaporation were the dominant natural hydrochemical mechanisms. On the one hand, water–rock interaction accounted for 71.6% and 75.1% of the total hydrochemical variations of surface water and groundwater, respectively. Of these, sulfate and chloride weathering explained 46.7% and 55.3%, respectively, while carbonate weathering explained 25.0% and 19.9%, respectively, of the total hydrochemical variations in surface water and groundwater. On the other hand, human activities explained 12.9% and 14.2% of these hydrochemical variations, respectively. Ion exchange was also determined as an important hydrochemical process. For stream water, livestock farming was found to be the most important artificial pollution source, whereas for groundwater in the plains, irrigation was the major contributor.

**Author Contributions:** Conceptualization, H.C.; data curation, D.L.; formal analysis, D.L.; funding acquisition, S.J.; investigation, H.C.; methodology, D.L.; project administration, A.L.; resources, A.L.; software, H.C.; supervision, S.J.; validation, D.L.; visualization, H.C.; writing–original draft, H.C.; writing–review and editing, D.L. All authors have read and agreed to the published version of the manuscript.

**Funding:** The research is supported by the National Natural Science Foundation of China (41630859, 41671026, 41701024) and Major Project of the Science and Technology of Qinghai, China (2019-SF-A4-1).

**Conflicts of Interest:** The authors declare no conflicts of interest.

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
