# Peer review of "Possible Hydrochemical Processes Influencing Dissolved Solids in Surface Water and Groundwater of the Kaidu River Basin, Northwest China"

_water, doi:10.3390/w12020467_

Round 1

Reviewer 1 Report

The authors have evaluated the surface water and groundwater in the Kaidu river basin, China. This is an important issue to be addressed especially in regions with water demand. This manuscript presents an interesting study in this field and its findings might be useful to other studies aimed at similar topics. However, the manuscript reserves more scope for improvement. I elaborate my comments below.

- L15, TDS abbreviation may be removed as it not used in the abstract. Similarly PCA.

- L15-16 and 18-19, both are repetition of the information on rock-water interaction

- it would be useful to include a sentence on the minerals that weather and dissolve in the water

-22-24 – better id the minerals that lead to SO4, Cl and Carbonate weathering are pointed out.

- please mention the time of sampling in the abstract

- line 25, remove ‘change’

-line 30, change to ‘Arid and semi-arid areas always suffer from water…’

- L 41, reference style should be changed

- Introduction is well-written, but the location of many places mentioned will not be clear to international readers. Suggest to include the country name when a river basin is mentioned.

- also, there is mix up of several regions of the world in in the introduction. Would suggest to start globally, then narrow to asia and then china.

-1-2 sentences about the importance of this study in the Kaidu river basin including the present water needs, supply and demand, any pollution issues, how much of the water demand is met from surface water and how much from groundwater etc. can be included. This will strengthen the need for this study.

- section 2 has not citations. Has all this data been collected by the authors?

-L88-97, geology should come after the paragraph on the climate.

Sequencing of paragraphs in introductions and study area makes it bit difficult to follow.  Would suggest to refer to these manuscripts as examples for the same:

https://doi.org/10.1007/s12665-019-8479-6

 https://doi.org/10.1007/s12665-011-1368-2

- L 118, please change to 3. Data and methods

-normally, the hydrogeochemical processes are first identified and then the SI of minerals are calculated to support the processes identified. Here the authors have used the other way around. I am not suggesting to change it, but request the authors to consider my comment.

- in the results section the authors have combined all the results i.e. surface water (SW) and groundwater (GW) and reported the results. In my opinion this is not the right method as they have completely different processes controlling their characteristics. Also, SW influence the GW composition. I would suggest that the authors represent the results separately.  

- L170, how was TDS measured? This information is not available in the methodology.

- how did the authors ensure the accuracy of the methods and the precision of the results? This should also be included in section 3

- L 205 to 209, the minerals that are over or under saturated in the water samples are reported. But what does that mean? What are the processes leading to these SIs?

- hope the SI is not calculated with SW and GW put together!

-The interaction between SW and GW would be interesting to study, which is not presented in detail in the manuscript. Please refer to https://doi.org/10.1007/s13201-013-0138-6

- L191-192, 200-201. If Ca-HCO3 is the dominant groundwater type, then how Na and Cl will be high in groundwater all the time. This is contradictory.

- L291, Estimation of the contributing sources

Please proofread the manuscript before submitting to avoid any unexpected typos or grammatical errors.

Author Response

Dear editors and reviewers,

On behalf of my co-authors, we thank you very much for giving suggestions to our manuscript, we appreciate the editors and reviewers very much for your positive and constructive comments and suggestions on our manuscript entitled “Possible hydrochemical processes influencing dissolved solids in surface water and groundwater of the Kaidu River Basin, northwest China” (ID: water-670939).

We have studied the comments carefully and have made revision highlighted. We have tried our best to revise our manuscript according to the comments. The main corrections in the paper and the responds to the reviewer’s comments are as flowing:

Responses to Reviewer #1’s Comments

[1] L15, TDS abbreviation may be removed as it not used in the abstract. Similarly PCA.

Answers: Thank you for your suggestion. We have removed the abbreviation of TDS and PCA in the abstract.

[2] L15-16 and 18-19, both are repetition of the information on rock-water interaction. it would be useful to include a sentence on the minerals that weather and dissolve in the water.

Answers: Thank you for your suggestion. Firstly, we removed the repetition. The major minerals that would dissolve into water was included in the abstract. They are highlighted in the manuscript with yellow. More information can be seen in line 20-21.

Lines 20-21: Sulfate, chloride and carbonate weathering were the major water- rock interaction processes.

[3] 22-24 – better add the minerals that lead to SO4, Cl and Carbonate weathering are pointed out.

Answers: Thank you for your suggestion. The abstract included the minerals that lead to high SO4, Cl and Carbonate weathering. They are highlighted in the manuscript with yellow. More information can be seen in line 20-21.

[4] please mention the time of sampling in the abstract

Answers: Thank you for your suggestion. We added the sampling time in the abstract in lines 12-13.

Lines 12-13: 109 surface water samples and 129 groundwater samples collected during August 2015 to September 2016…

[5] line 25, remove ‘change’

Answers: Thank you for your suggestion. We removed ‘change trend’ in line 25 in the former manuscript.

[6] line 30, change to ‘Arid and semi-arid areas always suffer from water…’

Answers: Thank you for your suggestion. We changed the ‘semiarid’ in the former manuscript into ‘semi-arid’ in this manuscript in line 28.

[7] L 41, reference style should be changed

Answers: Thank you for your suggestion. We changed the reference style in this manuscript. They are highlighted in the manuscript with blue font. More information can be seen in line 37, 45, 49, 141, 285, 286.

[8] Introduction is well-written, but the location of many places mentioned will not be clear to international readers. Suggest to include the country name when a river basin is mentioned. also, there is mix up of several regions of the world in in the introduction. Would suggest to start globally, then narrow to asia and then china.

Answers: Thank you for your suggestion. We added more information of the regions mentioned in the manuscript. They are highlighted in the manuscript with blue font. More information can be seen in line 38, 40, 294, 296, 299.

[9] 1-2 sentences about the importance of this study in the Kaidu river basin including the present water needs, supply and demand, any pollution issues, how much of the water demand is met from surface water and how much from groundwater etc. can be included. This will strengthen the need for this study.

Answers: Thank you for your suggestion. We improved this information in this manuscript with a paragraph in lines 56-65.

Lines 56-65: The Kaidu River is the major recharge source of the Bosten Lake, which is the largest inland freshwater lake in China (with water area of 1210.5 km2 and storage of 73.03×108 km3 when the lake elevation is 1047m above sea level (a.s.l.)). The Kaidu River is the major water resource for people life and economic development in Bayingolin Mongol Autonomous Prefecture (Bazhou), as well as for the ecological construction and environmental protection in the Yanqi Basin, the Konqi River Basin and the lower reaches of the Tarim River Basin. Security of water resource in the Kaidu River Basin is of great importance to regional sustainable development. Hydrochemical processes is a key factor that influence the water quality safety. What’s more, in recent years, with the rapid developing of irrigation agriculture and animal husbandry, a large amount of remnant charged into soil and water, threaten the security of water resources.

[10] section 2 has not citations. Has all this data been collected by the authors?

Answers: Thank you for your reminding. All this information was collected by us.

[11] L88-97, geology should come after the paragraph on the climate. Sequencing of paragraphs in introductions and study area makes it bit difficult to follow. Would suggest to refer to these manuscripts as examples for the same: https://doi.org/10.1007/s12665-019-8479-6; https://doi.org/10.1007/s12665-011-1368-2

Answers: Thank you for your suggestion. We adjusted the sequence of the paragraphs about climate and geology in lines 79-91.

[12] L 118, please change to 3. Data and methods

Answers: Thank you for your suggestion. We have modified the title of section ‘3 Data and method’ to ‘3 Data and methods’ in line 105.

[13] normally, the hydrogeochemical processes are first identified and then the SI of minerals are calculated to support the processes identified. Here the authors have used the other way around. I am not suggesting to change it, but request the authors to consider my comment.

Answers: Thank you for your suggestion. We did first identified the hydrogeochemical processes. Than the SI was used to support the results. They are highlighted in the manuscript with yellow. More information of SI can be seen in line 226-234.

Lines 226-234: The saturation indices indicate that the water in the Kaidu River Basin were saturated with aragonite, calcite and dolomite, while unsaturated with anhydrite, gypsum and halite. The calculated SI values of aragonite, calcite and dolomite ranged from -0.97 to 0.72, from -0.81 to 0.87 and from -2.53 to 1.64, with an average of -0.04, 0.12 and -0.18 respectively (Fig. 7), indicating that the aragonite, calcite and dolomite are saturated. They would precipitate under conditions that the carbon dioxide content decrease or temperature decrease. However, the SI values of anhydrite, gypsum and halite ranged from -3.80 to -1.20, from -3.55 to -0.95 and from -5.94 to -10.67, with an average of -2.23, -1.98 and -7.91 (Fig. 7), respectively, indicating the undersaturation of these minerals. They may dissolve continually under suitable conditions.

[14] in the results section the authors have combined all the results i.e. surface water (SW) and groundwater (GW) and reported the results. In my opinion this is not the right method as they have completely different processes controlling their characteristics. Also, SW influence the GW composition. I would suggest that the authors represent the results separately.

Answers: Thank you for your suggestion. In this study, we want to compare the hydrogchemical processes of surface water and groundwater in the Kaiddu River Basin, so we analyzed them together.

[15] L170, how was TDS measured? This information is not available in the methodology.

Answers: Thank you for your suggestion. We added the method we used to measure the TDS in section 3.1 in lines 128-129.

Lines 128-129: TDS was measured with wighing method.

[16] how did the authors ensure the accuracy of the methods and the precision of the results? This should also be included in section 3

Answers: Thank you for your suggestion. The detailed processes of sampling were shown in the manuscript in lines 118-124. The methods we used to measure the parameters were also shown in the manuscript in lines 125-129. The precision of the parameters was also shown in line 129.

[17] L 205 to 209, the minerals that are over or under saturated in the water samples are reported. But what does that mean? What are the processes leading to these SIs?

Answers: Thank you for your suggestion. What does over or under saturation mean? What are the processes leading to these Sis? About these questions, the detailed information is shown in lines 227-234. They are highlighted in the manuscript with yellow.

Lines 227-234: The calculated SI values of aragonite, calcite and dolomite ranged from -0.97 to 0.72, from -0.81 to 0.87 and from -2.53 to 1.64, with an average of -0.04, 0.12 and -0.18 respectively (Fig. 7), indicating that the aragonite, calcite and dolomite are saturated. They would precipitate under conditions that the carbon dioxide content decrease or temperature decrease. However, the SI values of anhydrite, gypsum and halite ranged from -3.80 to -1.20, from -3.55 to -0.95 and from -5.94 to -10.67, with an average of -2.23, -1.98 and -7.91 (Fig. 7), respectively, indicating the undersaturation of these minerals. They may dissolve continually under suitable conditions.

[18] hope the SI is not calculated with SW and GW put together!

Answers: Thank you for your suggestion. The SI of surface water and groundwater did not calculated together.

[19] The interaction between SW and GW would be interesting to study, which is not presented in detail in the manuscript. Please refer to https://doi.org/10.1007/s13201-013-0138-6

Answers: Thank you for your suggestion. We did not present the interaction between surface water and groundwater in detail in this manuscript, but we would do more research about the interactions in the future. We think that this question can be explained more exactly with stable oxygen and hydrogen isotopes.

[20] L191-192, 200-201. If Ca-HCO3 is the dominant groundwater type, then how Na and Cl will be high in groundwater all the time. This is contradictory.

Answers: Thank you for your suggestion. It is not contradictory. The reasons are as following. First, the predominant hydrochemical types of the plain groundwater are Ca2+- HCO3- and mixed type water. The proportion of mixed type of water is very high. Second, “the concentration of Na+ and Cl- in groundwater is high all year round”, it does not mean that “the concentration of Na+ and Cl- in groundwater is the highest among all ions”.

[21] L291, Estimation of the contributing sources

Answers: Thank you for your suggestion. We have changed the title of section 4.4 into “4.4 Estimation of the contribution sources”, see in line 262.

[22] Please proofread the manuscript before submitting to avoid any unexpected typos or grammatical errors.

Answers: Thank you for your suggestion. We did it.

Reviewer 2 Report

Paper name:

- paper name should be perhaps changed into “Possible hydrochemical processes influencing dissolved solids in surface water and groundwater of the Kaidu River Basin, northwest China” or something similar to expresses the paper content.

Abstract:

- in 169 words, describes the paper content in the desired format but contains a misleading information that 235 groups of water were under the scope of view – “groups” should be changed to “samples”.

Key words:

- keywords (stream water; groundwater; Kaidu River Basin; hydrochemical process; water-rock interaction; human activity) are adequately selected;

Text:

- references and citations are absolutely matching, but there are few references that are cited differently: Wang et al. (2018) instead of [7], Shi et al. (2018) instead of [3], Li et al. (2018) instead of [10], Schoeler (1965) instead of [21], Chiogna et al. (2018) instead of [40];

- some facts are repeated from the introduction also in chapter 2. Study site in lines 82-87;

- lines 95-97: what is the meaning of moisture content of deposits? Is it permeability / groundwater resources / groundwater content / hydrogeological productivity? Moisture content is usually attributed to soils, but not deposits / rocks;

- lines 109-112: formulations about groundwater depth increase and decrease are perhaps contradictory – it would be better to use “groundwater level” instead, and then is relative change is more simple to describe;

- line 150 and further: numbering of equations starts with No. 6 (!), when describing formulas with cations, please show also the input unit (mg/l, mmol/l, meq%) to make the results more reproducible;

- lines 246-250: please, formally adopt reactions / equations;

- lines 284-290 and 339-346: the formulations in the whole paragraph should be checked (improper sentence beginning with prepositions as “while” etc.);

- lines 348-363 (Conclusion): perhaps two decimal places in numbers are too exact/precise in comparison to the analysed material and PCA results, one decimal place (if any) should be enough – this should be perhaps applied also for the abstract;

- the whole paper contains too little information about lithology / chemical composition of rocks in the catchment, what should be necessarily added to the chapter describing the study site, also adding some general geochemical/lithological map would be valuable in addition to Fig. 1.

Figures:

- Figures 2 – 8 contain more information than is described in the text, what is worth of appreciation. Still, in Fig. 3 there are too many symbols for sampling seasons and sample sources – Fig. 3a and 3b should be separated and the legend explaining water types 1-9 and A-G should be left one for both perhaps in the figure caption, the same also for Fig. 5;

Tables:

- tables are adequately accompanying the text, perhaps their format (font size) should be inspected.

Author Response

Dear editors and reviewers,

On behalf of my co-authors, we thank you very much for giving suggestions to our manuscript, we appreciate the editors and reviewers very much for your positive and constructive comments and suggestions on our manuscript entitled “Possible hydrochemical processes influencing dissolved solids in surface water and groundwater of the Kaidu River Basin, northwest China” (ID: water-670939).

We have studied the comments carefully and have made revision highlighted. We have tried our best to revise our manuscript according to the comments. The main corrections in the paper and the responds to the reviewer’s comments are as flowing:

Responses to Reviewer #2’s Comments

Paper name

[1] paper name should be perhaps changed into “Possible hydrochemical processes influencing dissolved solids in surface water and groundwater of the Kaidu River Basin, northwest China” or something similar to expresses the paper content.

Answers: Thank you for your suggestion. We have changed the paper title into “Possible hydrochemical processes influencing dissolved solids in surface water and groundwater of the Kaidu River Basin, northwest China”.

Abstract

[1] in 169 words, describes the paper content in the desired format but contains a misleading information that 235 groups of water were under the scope of view – “groups” should be changed to “samples”.

Answers: Thank you for your reminding. We have changed “groups” into “samples” in line 12.

Line 12: … 109 surface water samples and 129 groundwater samples…

Key words

[1] keywords (stream water; groundwater; Kaidu River Basin; hydrochemical process; water-rock interaction; human activity) are adequately selected;

Answers: Thank you for your affirmation. We really appreciate your support.

Text

[1] references and citations are absolutely matching, but there are few references that are cited differently: Wang et al. (2018) instead of [7], Shi et al. (2018) instead of [3], Li et al. (2018) instead of [10], Schoeler (1965) instead of [21], Chiogna et al. (2018) instead of [40];

Answers: Thank you for your suggestion. We changed the reference style in this manuscript. They are highlighted in the manuscript with blue font. More information can be seen in line 37, 45, 49, 141, 285, 286.

[2] some facts are repeated from the introduction also in chapter 2. Study site in lines 82-87;

Answers: Thank you for your reminding. We have rewrote these two paragraphs in lines 56-65 and line 72-78. They have been highlighted with blue font.

Lines 56-65: The Kaidu River is the major recharge source of the Bosten Lake, which is the largest inland freshwater lake in China (with water area of 1210.5 km2 and storage of 73.03×108 km3 when the lake elevation is 1047m above sea level (a.s.l.)). The Kaidu River is the major water resource for people life and economic development in Bayingolin Mongol Autonomous Prefecture (Bazhou), as well as for the ecological construction and environmental protection in the Yanqi Basin, the Konqi River Basin and the lower reaches of the Tarim River Basin. Security of water resource in the Kaidu River Basin is of great importance to regional sustainable development. Hydrochemical processes is a key factor that influence the water quality safety. What’s more, in recent years, with the rapid developing of irrigation agriculture and animal husbandry, a large amount of remnant charged into soil and water, threaten the security of water resources.

Lines 72-78: The Kaidu River Basin is located at the southern slope of the Tianshan Mountains and the northern rim of the Tarim River Basin (Fig. 1). It covers an area of 44,147 km2, with the altitude ranges from 928 m above sea level (a.s.l.) to 4,796 m a.s.l.. The Kaidu River originates from glaciers in the Tianshan Mountains, runs through the small Yourdusi Basin, the big Yourdusi Basin and then a long mountain valley to the Dashankou hydrometric station, through the Yanqi Basin and finally ends up at the Bosten Lake. It is one of the largest rivers at the southern slope of the Tianshans (with mean annual discharge of 108.42 m3/s) [15].

[3] lines 95-97: what is the meaning of moisture content of deposits? Is it permeability / groundwater resources / groundwater content / hydrogeological productivity? Moisture content is usually attributed to soils, but not deposits / rocks;

Answers: Thank you for your reminding. We have rewrote this sentences in line 91.

The groundwater resource is more rich in the mountainous areas than the plain areas.

[4] lines 109-112: formulations about groundwater depth increase and decrease are perhaps contradictory – it would be better to use “groundwater level” instead, and then is relative change is more simple to describe;

Answers: Thank you for your reminding. We have rewrote this sentences in lines 95-100.

Line 95-100: The annual mean groundwater level ranges from 3 m to more than 28 m below the ground. In recent years, the groundwater level has decreased significantly. On average it has decreased 5 m to 14 m. The groundwater level is the shallowest in winter and deepest during growing season (June to August) and during autumn irrigation season (October). In regions with large area of farmland, the groundwater level is deeper than other areas.

[5] line 150 and further: numbering of equations starts with No. 6 (!), when describing formulas with cations, please show also the input unit (mg/l, mmol/l, meq%) to make the results more reproducible;

Answers: Thank you for your suggestion. We have added the units. More details can be seen in line 137, lines 147-148.

[6] lines 246-250: please, formally adopt reactions / equations;

Answers: Thank you for your reminding. We have formally adopt the reactions in lines 221-225.

[7] lines 284-290 and 339-346: the formulations in the whole paragraph should be checked (improper sentence beginning with prepositions as “while” etc.);

Answers: Thank you for your reminding. We have rewrote these two paragraphs. They have been highlighted with yellow. More details can be seen in lines 256-261 and lines 305-311.

Lines 256-261: The population in Yanqi oasis had increased before 2010 and then decreased. The farmland area, orchard area, number of livestock, the used pesticide and chemical fertilizer has increased significantly (Table 3). Pesticides and fertilizer residual in soil would not charge into groundwater or river in arid areas if without extreme events like heavy rainstorm. But they are potential menace for regional water resource security [4,37]. However, the animal husbandry, which is mainly distributed in water source areas, can contaminate the water directly.

Lines 305-311: Nomadic herding is highly developed in the mountain areas of the study area. In the downstream of the catchment, irrigation agriculture is widely distributed. In recent years, the population of livestocks, the irrigation area, the amount of pesticide and fertilizer used have raised significantly (Table 3). A large amount of chemical elements like nitrogen, phosphorous, and potassium charged into soil and water. They would change the hydrochemical processes, threaten the regional water source quality, enhance the spatial and temporal variance of hydrochemical characters. For example, the TDS of surface water and shallow groundwater would increase significantly after autumn irrigation.

[8] lines 348-363 (Conclusion): perhaps two decimal places in numbers are too exact/precise in comparison to the analysed material and PCA results, one decimal place (if any) should be enough – this should be perhaps applied also for the abstract;

Answers: Thank you for your reminding. In this manuscript, the results of PCA were shown with one decimal place. They are highlighted with blue font. The details can be fine in lines 19-20, lines 266-276, lines 319-323.

[9] the whole paper contains too little information about lithology / chemical composition of rocks in the catchment, what should be necessarily added to the chapter describing the study site, also adding some general geochemical/lithological map would be valuable in addition to Fig. 1.

Answers: Thank you for your reminding. We did want to add more information about lithology / chemical composition of rocks in the catchment. We have consulted a lot of data, paper and other materials. But we did not fine reliable information about this question. So we would do more work in this data scare area in the future.

Figures

[1] Figures 2 – 8 contain more information than is described in the text, what is worth of appreciation. Still, in Fig. 3 there are too many symbols for sampling seasons and sample sources – Fig. 3a and 3b should be separated and the legend explaining water types 1-9 and A-G should be left one for both perhaps in the figure caption, the same also for Fig. 5;

Answers: Thank you for your reminding. In this paper, we want to make more easy to compare the water types of stream water and groundwater, so we put them in one figure rather than separate it into two figures.

Tables

[1] tables are adequately accompanying the text, perhaps their format (font size) should be inspected.

Answers: Thank you for your reminding. We have checked the tables carefully.
